# Development of a Waste Management Strategy in a Steel Company

Ioana Fărcean [1,*], Gabriela Proștean [1], Erika Ardelean [2,*], Ana Socalici [2] and Marius Ardelean [2]

[1] Research Center for Engineering and Management (RCEM), Faculty of Management in Production and Transportation, University Politehnica Timisoara, Remus Street, No. 14, 300191 Timisoara, Romania; gabriela.prostean@upt.ro

[2] Faculty Engineering of Hunedoara, University Politehnica Timisoara, Revolutiei No. 5, 331128 Hunedoara, Romania; virginia.socalici@fih.upt.ro (A.S.); marius.ardelean@fih.upt.ro (M.A.)

* Correspondence: ioana.farcean@student.upt.ro (I.F.); erika.ardelean@fih.upt.ro (E.A.)

**Abstract:** The management of waste, especially ferrous waste, poses great problems in the steel industry due to strict regulations on preventing, reducing, or even eliminating the factors that generate a high degree of environmental pollution (landfills resulting from the steel industry and adjacent industries—mining, energy, etc.). The present paper presents a synthesis of the specialized literature regarding the processes used, both worldwide and nationally, regarding the transformation of raw materials (ores or concentrates) and iron-containing waste (steel mill dust, mill scale and scale, sludge from agglomeration factories, sideritic waste, etc.) into by-products that can be used in the steel industry. For technological reasons, the option of pelletizing powdered waste was applied—in laboratory conditions, according to its own recipes, with results that justify the application of the technology on an industrial scale (appearance after hardening; drop resistance). The aim of the paper was to identify a practical solution; based on this solution, original conceptual models of organizational strategies (management and processing, respectively, recovery of ferrous waste within steel companies) were developed, such as a concentration strategy, diversification, vertical integration, etc. Within graphical representations of the proposed strategies, other processing variants were mentioned: agglomeration; briquetting.

**Keywords:** waste management; steel industry; organizational strategy; ferrous scrap; conceptual model of organizational strategy

## 1. Introduction

Significant amounts of manufacturing waste (steel mill dust, scale, slag, sludge, etc.) are generated daily in the steel industry; they have an intrinsic value determined by their content of useful elements (Fe, C, etc.). For these reasons, and due to the legislative rules in force on the reduction in pollution caused by industrial waste landfills, it is necessary to recover these types of waste by applying the circular economy concept.

Within the steel industry, the concept of a circular economy promotes a circular production flow, in which the main objective is to recycle/reuse/recover production waste as secondary raw materials in the processes in which they were generated [1–4].

The definitions of waste and secondary raw materials will always fluctuate according to technological performance and economic interests [5], always representing a problem of the utmost importance that will lead to the emergence of numerous discrepancies; through processing, waste can become a secondary raw material that is as valuable as any raw material from a natural source (iron ore, coal, etc.).

Figure 1 presents the idea that waste is the set of substances, materials, and residues generated by industrial activity and for which there is the possibility of elimination from the production cycle or reintroduction (an option recommended for steel companies generating

waste with a high content of useful elements), which can only be identified through appropriate management, based on the application of the concept of circular economy.

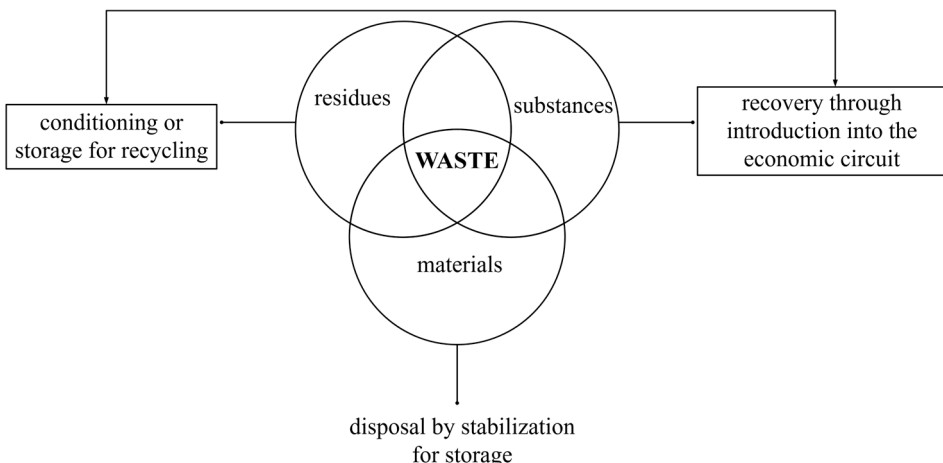

**Figure 1.** Graphical representation of the notion of waste.

Worldwide, the estimated amount of waste from the steel industry is 400 million tons/year. According to estimates, the amount of waste generated in a single year by the Romanian steel industry is 4.8 million tons/year. Only 33% of the total production waste generated in Romania [6] has been recovered or recycled and recovered, and most of the waste is being disposed of in nature on the disused sites of major companies in the steel industry, the mining industry, the chemical industry, the energy industry, etc. (ponds for the settlement of siderite waste from siderite mining, located in Hunedoara county; slag dumps in Galați and Reșița; pyrite ash dumps in Valea Călugărească; red mud dumps from the processing of bauxite ore in Oradea, etc.).

The Environmental Report for Romania's Energy Strategy 2020–2030, containing prospects for 2050, presents information on both current and historical industrial waste landfills, which occupy about 844 ha and 360 ha of land [6,7], respectively; these are in counties where the mining and steel industries have a history and/or are ongoing.

In the practices of the steel industry worldwide, the most common iron-containing wastes are the following: sinter dust and sludge; blast furnace dust and sludge; converter dust and sludge; steel mill dust (from Siemens Martin steel mill); electrofilter dust from the electric steel mill; scale and mill scale sludge; ferrous fraction of steel mill slag [4,8–16].

Following a review of the literature [11–15], the estimated amounts of steel scrap from different Fe-C alloy production streams were identified. These are presented in Table 1.

**Table 1.** Estimated average quantities of waste generated [11–15].

| Waste Type | Sector of Provenance | Estimated Average Quantities per Tonne of Steel/Cast Iron |
|---|---|---|
| Steel mill dust | Steelmaking | 10–25 kg |
| Blast furnace dust | Cast iron elaboration | 10–50 kg |
| Slag | Cast iron/steel elaboration | 250 kg |
| Converter dust | Steelmaking | 13–20 kg |

Referring to the policies and measures to be implemented based on the rules laid down in legislative documents (European Green Pact, Action Plan on the Circular Economy for Romania Project, Directive 1999/31/EC on landfills, Directive (EU) 2018/850 on waste, etc.), the European Union has set an ambitious target for only 10% of all residual waste to be disposed of in landfill by 2035 [9].

The production waste generated in iron and steel making processes (blast furnace dust, steel mill dust, scale, etc.) must be reused in the context of the circular economy and transformed into by-products (pellets, agglomerates, sponge iron, etc.), which will form part of the raw material and material base of steel companies, thus strengthening the sustainable development of the steel industry. Applying the above approach contributes to minimizing landfill waste, reducing gas emissions, and conserving primary raw material resources [16].

## 2. Technology Overview

The exploitation of the Fe content of ferrous ores, by exposing them to temperatures below the melting temperature of iron, is achieved through the direct reduction process, which involves the transformation of iron ore into metallic iron (solid state) [11,17,18].

Worldwide, direct reduction processes result in iron sponge called DRI (direct-reduced iron) or CDRI (cold-direct-reduced iron), and HDRI (hot-direct-reduced iron) which is then hot-compacted and turned into briquettes, called HBI (hot-briquetted iron [11,17,19–24].

In industrial practices, according to the literature [11,25–27], there are many types of direct reduction processes, classified according to various criteria; Figure 2 lists most of the direct reduction processes that have appeared worldwide.

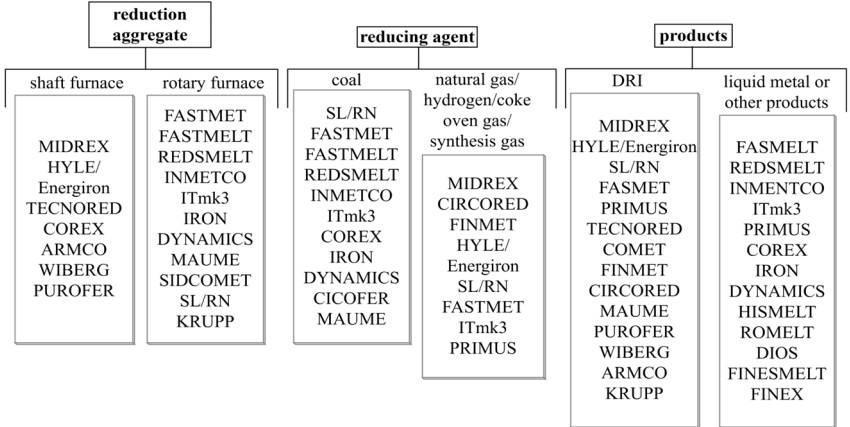

**Figure 2.** Typology of direct reduction processes, according to [11,25–27].

According to [11,17,20,22], iron sponge can be produced in various forms for use in electric arc furnaces, induction furnaces, oxygen converters, and blast furnace charges as a secondary raw material if the following chemical composition conditions are met: Fe content—min. 90%; $Al_2O_3$ + MgO—max. 2%; basicity—0.10–0.20; C content—less than 2%.

In the following, the most-used iron ore reduction processes that are practiced worldwide are briefly presented.

1. The Midrex process.

In the Midrex process, iron oxides are reduced from ore (in the form of pellets and lumps), whose minimum iron content is 67% [28], in a shaft furnace reduction plant.

The direct reduction plant used in the Midrex process can operate on natural gas, coal combustion syngas, or coal oven gas, with the fuel being used as the reducing agent [28].

The iron sponge manufactured by the Midrex process comes in different shapes and sizes:

- CDRI contains, on average, 90–94% Fe, is cooled to ambient temperature, and is basically used as a charge in electric arc furnaces [11].
- Conventionally, DRI is cooled before being discharged from the direct reduction plant, but technical modifications have been implemented for the discharge of hot DRI (HDRI) [29].
- HBI contains the same percentage of iron as the CDRI and HDRI variants; in this case, the iron sponge is discharged hot from the furnace into a press that shapes the reduced

material into briquettes, which are then used in electric arc furnaces, blast furnaces, and oxygen converters [11,28].

Today, the popularity of HBI as an ore-based product is growing, becoming a key element in the environmental strategies of several traditional integrated steel mills, as well as a valuable by-product for steel producers [30].

The technology on which this process is based is considered the most widely used and successful iron-sponge-manufacturing technology.

2.  The HYL/ENERGIRON process.

In the HYL process, oxygen from iron ore (initially processed as pellets or lumps) is removed in a retort/reactor plant using reducing gases obtained from methane [11,26,31].

ENERGIRON is HYL's innovative direct reduction technology that is designed to use various types of reducing gas sources or pure $H_2$ to reduce iron ores to metallic iron for use in smelters. One of the main advantages of the HYL process is the configuration based on the independent reduction in the generation and gas reduction section. The HYL process offers great flexibility in the use of alternative gas reduction sources [31].

3.  SL/RN—rotary kiln reduction process.

The process consists of charging a rotary kiln with iron ore in lump or pellet form, solid reducing agent (coal), and desulfurizing agent (lime or dolomite) [11,19,32,33].

Compared to direct reduction shaft furnaces (Midrex, HYL processes), rotary furnaces are characterized by greater flexibility in iron ore, high-energy consumption, and low productivity [34].

As a solution to dwindling ore resources and the availability of increasing amounts of ores and fine concentrates, Metso Outotec (the company that owns the SL/RN technology) has developed a variant of the direct reduction process called SL/RN-Xtra. This new adaptation of the classic SL/RN process allows the production of DRI directly from concentrates in a single plant, combining pelletizing and direct reduction processes [33].

4.  The ITmk3 process.

This process involves making a mixture of iron ore/fine ore concentrates and unburnable coal to obtain raw pellets, which are charged into the rotary hearth furnace and transformed by simultaneous reduction and melting into granulated iron called "iron nuggets" [26,35–41].

The steps taken in the ITmk3 process are to make pellets, reduce and melt the pellets in the temperature range 1350–1450 °C, a process that lasts about 10 min, separate metallic iron (granulated iron) from the slag, treat the flue gases, and recover heat [36,39,40,42,43].

Granulated iron (iron nuggets) manufactured by the ITmk3 process has a low carbon content, does not contain impurities, and represents a substitute for cast iron because it has similar or even superior physical and chemical properties.

The ITmk3 process is different from the other processes because the reduction and slag separation operations take place simultaneously or in very short time intervals [40].

The SL/RN and ITmk3 processes represent outstanding solutions for the processing of iron ores and by-products containing iron, to produce by-products intended for the elaboration processes of metal alloys (cast iron, steel) [37].

In essence, we highlight that the direct reduction process is applicable not only to iron ores, but also to iron ore waste that was previously processed by means of classic processes (pelletizing, briquetting). This aspect is supported by the research presented in the national specialized literature, which constituted a starting point for the research and experiments undertaken.

Taking into account the processes presented and the results obtained by different manufacturers worldwide, and based on the literature review [11,31,34,44–46], process efficiency data are summarized in Table 2. Based on cost–benefit analyses, each manufacturer locally implements the technological solution that allows them to achieve the proposed objectives.

**Table 2.** Comparative analysis of direct reduction processes [11,31,34,44–46].

| Process | Degree of Metallization | Benefits |
| --- | --- | --- |
| Midrex | 92–97% | High productivity; low consumption of electricity and refractory material. |
| HYL/ENERGIRON | 92–95% | Flexibility in the use of alternative sources of gas reduction. |
| SL/RN | 85–95% | Flexibility in terms of raw materials. |
| ITmk3 | 96–97% | Low energy consumption; reduced gas emissions compared to those resulting from the production of cast iron in the furnace. |

5. The state of research on the direct reduction process in Romania.

This part of the paper discusses aspects of the current state of research at the national level. Research carried out by a team of specialists from the Polytechnic University of Bucharest, Faculty of Engineering Hunedoara, and representatives of a steel company, aimed to establish optimal technological flows for the recovery of ferrous steel waste within a direct reduction plant of a rotary furnace [14,25,47–49].

The unconventional technology developed by the research team involved the introduction of inexpensive raw materials (ore powder, sludge, etc.) into the furnace, together with coke or coal powder (reducing agent) and a dispersing agent that also served as a thermal agent (ceramic balls with a diameter of 12 mm and triangular steel refractory steel triangular prisms) [49].

The products generated in the direct reduction process were ferrous metal powder, combustion gases, and sterile. After the reduction process, the iron powder obtained was separated from the residue of the reducing agent through magnetic concentration [25,50].

In conclusion, the results of the research conducted by UPB-CEMS specialists were represented by obtaining an iron powder containing a minimum of 85% metallic iron and valorizing the product obtained in the steel industry [25]; the results represented an important contribution to the field of research surrounding the recovery of ferrous industrial waste. The authors considered it imperative to emphasize the originality of the development of this technology in Romania.

Although the industrial-scale implementation of unconventional technology would have been considered advantageous for the Romanian steel industry, the installation remained only in the pilot phase.

Within a research project PN2 no. 31-098/2007 [4], contracts for teaching staff of the Faculty of Engineering in Hunedoara have been made from ferrous waste powdery micro-pellets/pellets. The basic idea of the research was to obtain mixed pellets with iron and carbon content, which were later sintered. The main processed waste included the following: steel mill dust, agglomerate furnace sludge, graphite electrode scraps, and bentonite, which was used as a binder.

The pellet samples were introduced into the interior of the rotary tube furnace (experimental setup); the direct reduction and metallization process of the pellets was considered to be complete when the pilot flame could no longer be maintained, resulting in pre-reduced pellets, similar in composition to sponge iron (degree of metallization ranging from 84 to 92%). The main result of interest was the obtained degree of metallization of the sponge iron, which favors the use of the product in casting and steelmaking processes [4,14,48,51].

The totality of the aspects presented above represented the starting point of our own research conducted under laboratory conditions, which is detailed further in the following section.

## 3. Materials and Methods

Pellets, briquettes, and agglomerates (by-products) resulting from the processing of waste with a high iron content (scale, steel dust, blast furnace dust, etc.) are used in the charge of iron and steelmaking aggregates as secondary raw materials, partially or completely replacing iron ore or other raw materials [5,11,13,15,52,53].

The processing of waste through pelletization involves the formation of crude spheres by rolling a mixture of waste, binder (cement, bentonite), and a quantity of water in a pelletizer-type facility. The process of forming raw pellets occurs as a result of the adhesion of fine particles, which are entrained in a rotating motion, leading to the formation of cohesive forces created by the added liquid (water), which facilitates their formation [5,11,13,52–55].

A series of experiments for processing small and powdery waste with iron content through pelletization were carried out within the hall at the Faculty of Engineering, Hunedoara. A series of specific recipes were developed, in which a diverse range of steel waste, generated within current steel manufacturing flows (steel dust, scale, and mill scale sludge, etc.), and recovered from historical waste deposits containing iron, located within Hunedoara County (sideritic waste), were used.

The chemical composition of the main waste materials included in the formulated recipes is presented in Table 3. Chemical composition was determined in a specialized laboratory within a steelmaking company, using a Thermo Scientific ARL PERFORM'X sequential X-ray fluorescence (XRF) spectrometer, with the Thermo Scientific UniQuant program employed for quantitative XRF analysis (manufactured by Thermo Fisher Scientific, Carlsbad, CA, USA) [56].

**Table 3.** Chemical composition of processed waste.

| Steel mill Dust, [%] | | | | | | |
|---|---|---|---|---|---|---|
| $Fe_2O_3$ | ZnO | CaO | $SiO_2$ | MnO | $SO_3$ | Other elements |
| 40.60 | 23.34 | 4.69 | 4.63 | 3.53 | 3.36 | 19.85 |
| Blast furnace agglomeration dust, [%] | | | | | | |
| $Fe_2O_3$ | $SiO_2$ | CaO | $Al_2O_3$ | MgO | ZnO | Other elements |
| 50.32 | 15.38 | 10.38 | 9.40 | 2.87 | 9.30 | 2.35 |
| Sideritic waste, [%] | | | | | | |
| $SiO_2$ | $Fe_2O_3$ | CaO | MgO | $Al_2O_3$ | Other elements | |
| 44.94 | 21.82 | 18.24 | 5.09 | 2.89 | 7.02 | |
| Sideritic waste concentrate, [%] | | | | | | |
| $SiO_2$ | $Fe_2O_3$ | CaO | MgO | $Al_2O_3$ | Other elements | |
| 35.53 | 24.67 | 18.22 | 9.96 | 5.64 | 5.98 | |
| Scale, [%] | | | | | | |
| Fe | C | Other elements | | Oil content | | |
| 79.63 | 7.98 | 12.39 | | 2.45 | | |
| Mill scale sludge, [%] | | | | | | |
| Fe | Other elements | | Water | | Oil content | |
| 95 | 5 | | 0.55 | | 0.77 | |

Based on the current state of research in the specialized literature [13,14,48,51,52] and depending on the particle size structure and chemical composition of the waste, 8 recipes were designed and realized. We note that, during the experiments, the amount of pelletizing mixture used varied in the range of 500–1000 g. Figure 3 shows images of some of the types of waste processed using pelletizing process technologies. The waste was classified and

ground, using only the corresponding fractions, according to the specialized literature (size smaller than 0.5 mm).

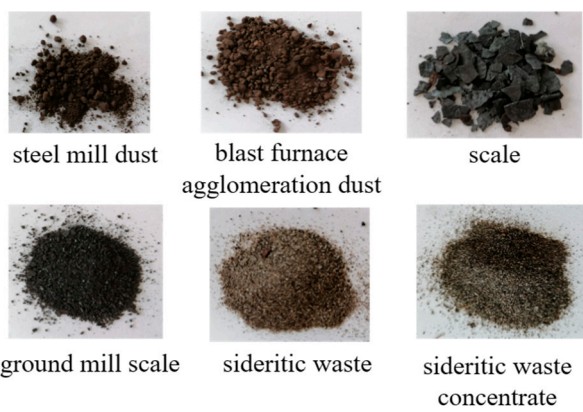

| steel mill dust | blast furnace agglomeration dust | scale |
| ground mill scale | sideritic waste | sideritic waste concentrate |

**Figure 3.** Some of the waste processed by pelletizing.

The waste pelletization process was carried out in a disc pelletizer with the following technical specifications: disc diameter—970 mm; disc height—90 mm; inclination angle—45°; 8 rot/min.

The binders used were lime, bentonite, and cement. To maintain the shape of the pellets, a negligible amount of graphite was added at the end. The obtained pellets are presented in Figure 4.

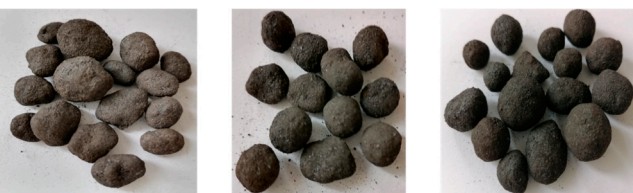

**Figure 4.** Pellets obtained from the pelletization of waste.

The pellet samples were sintered in an oven by placing them in crucibles on a graphite bed, covered with graphite to reduce iron oxides. The thermal regime of the process is presented succinctly in Figure 5.

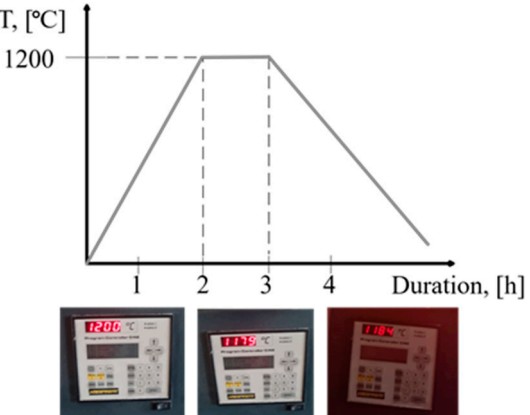

**Figure 5.** The sintering regime of pellets.

Figure 6 presents, synthetically, the stages of the sintering/reduction process of the obtained pellets.

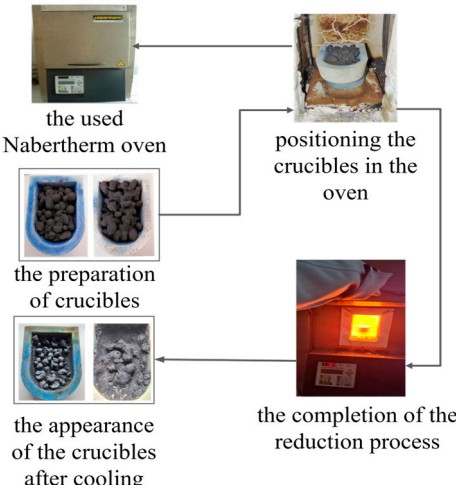

**Figure 6.** The stages of the process of reducing the obtained pellets.

After the pellets were sintered, a microscopic analysis was performed using a stereomicroscope, highlighting the following aspects: the shape of the sintered/reduced pellets and the appearance of Zn crystals/dendrites (see Figure 7).

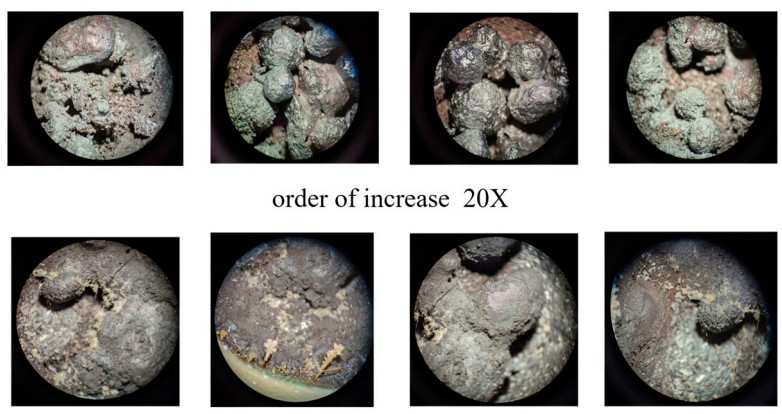

**Figure 7.** Microscopic analysis of sintered pellets.

For the qualitative determination of the pellets, drop resistance tests were performed for the different batches of recipes, resulting in an average percentage of 83% for the corresponding pellets.

The results obtained from preliminary experiments recommend this method of valorization of iron-containing waste for implementation in industrial practice, but with the adoption of technological corrections: monitoring the zinc content in steel dust, reducing the sintering temperature of pellets, and optimizing the chemical composition of recipes in terms of the waste used, correlated with the availability of these wastes.

## 4. Proposed Solutions

Any type of manufacturing company, especially steelmaking companies, needs a strategy to ensure that raw materials and materials resources are allocated efficiently, with a focus on the proper management of production waste. This aspect must be considered when major decisions are made regarding resource allocation or when diversifying activities (the steel company becomes a processor/recycler of steel waste also).

To develop conceptual models of organizational strategies, the authors laid the groundwork for designing an approach to strategic management associated with industrial waste

management activities; the specific steps to follow in this endeavor are presented in Figure 8.

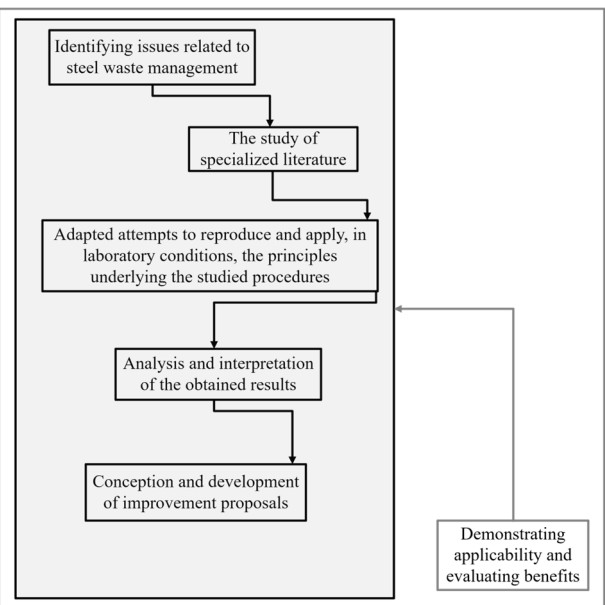

**Figure 8.** The stages of approaching strategic management in the process of designing an organizational strategy regarding industrial waste management.

The process of designing the approach consisted of a critical analysis of the specialized literature, highlighting the limitations of the technologies studied, and opting for the reproduction of a technological version of the pelletizing process, adapted to the conditions of a teaching laboratory. The results of the experiments that were carried out were optimal according to the literature and, as a result, a series of proposals was generated to improve waste management activities; this series takes the form of conceptual models of organizational strategies related to the specifics of steel companies. These are presented below.

A strategy establishes how the desired results will be achieved within the specific objectives; the objective is to increase production volume and consequently achieve substantial sales growth. The application of growth strategies is recommended, such as the concentration strategy, the vertical integration strategy, and the diversification strategy [57].

The conceptual models related to the strategies aim to expand the raw material base within steel companies. The models have been developed by the authors following a study of the literature and preliminary experimental research, as presented below.

### 4.1. Development of the Conceptual Model for the Concentration Strategy

For a steel company that faces problems regarding waste generation in current production flows, a concentration strategy can be applied, as shown in Figure 9.

By implementing the strategy of concentration within any type of company, the aim is to expand operations in the existing fields of activity. The implementation of a concentration strategy within any type of society, especially a steel company, allows the respective society to take advantage of its strengths in terms of the technical and managerial aspects it has or can acquire within a related field of its business activity.

In this case, the Fe-C alloy steelmaking company has the option to become a waste processing and valorization company, which would simultaneously lead to increased production activity, improved efficiency in waste management processes, and improved sustainability.

It is recommended that the concentration strategy be applied only when the company seeks to expand the volume of activity in its current business field, or, in the context of applying the concepts of sustainable development, when the company needs to reorient itself regarding raw materials or diversify the materials used in the production of its own products.

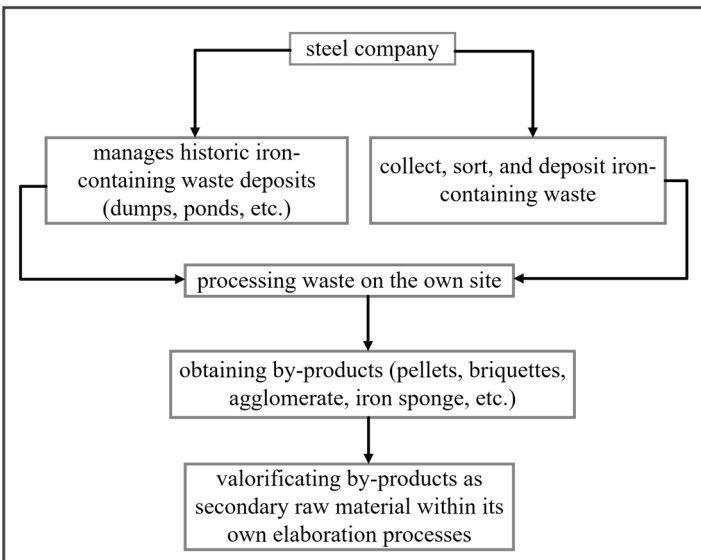

**Figure 9.** Conceptual model for the concentration strategy.

The main advantages of applying the conceptual model of the concentration strategy within a steelmaking company are highlighted by the following aspects:

- Creation of diverse proprietary resources of secondary raw materials;
- Expansion of the company's material base through waste production;
- Reducing the quantities of scrap metal that, in certain situations, can harm the quality of the semi-finished product/finished product;
- Generation of added value through the processing and valorization of stored waste, which contributes to the reduction in pollution caused by its long-term storage and aligning with environmental legislation standards.

It should also be considered that the competitive environment is almost non-existent regarding the processing of historical waste deposits, which are no longer produced.

The potential disadvantages that can arise from the application of the concentration strategy conceptual model within a steel company can be directly caused by the following factors: difficulty level of retrieving and transporting historically stored waste to the processor, the absence of a plan to green/combat the effects of landfilling on the environment, the lack of or difficult access to the necessary waste processing machinery and facilities, the limited quantity of historically stored steel waste that is no longer generated, possible changes in the manufacturing flow, and the organization's limited or insufficient financial resources.

### 4.2. Development of the Conceptual Model Applicable to a Joint-Venture Strategy

The specialized literature [57] provides three possibilities for implementing a concentration strategy: market development, product development, and horizontal integration.

In the steel industry, it is difficult to attract new beneficiaries if you do not already have a certain stability in the market. It has been observed that large companies tend to integrate horizontally by absorbing/acquiring small companies in the field, resulting in today's firms of the conglomerate type.

Figure 10 presents the conceptual model of a joint-venture strategy, a type of strategy based on an agreement between at least two companies that decide to work together to achieve a common goal of growth and prosperity for each partner.

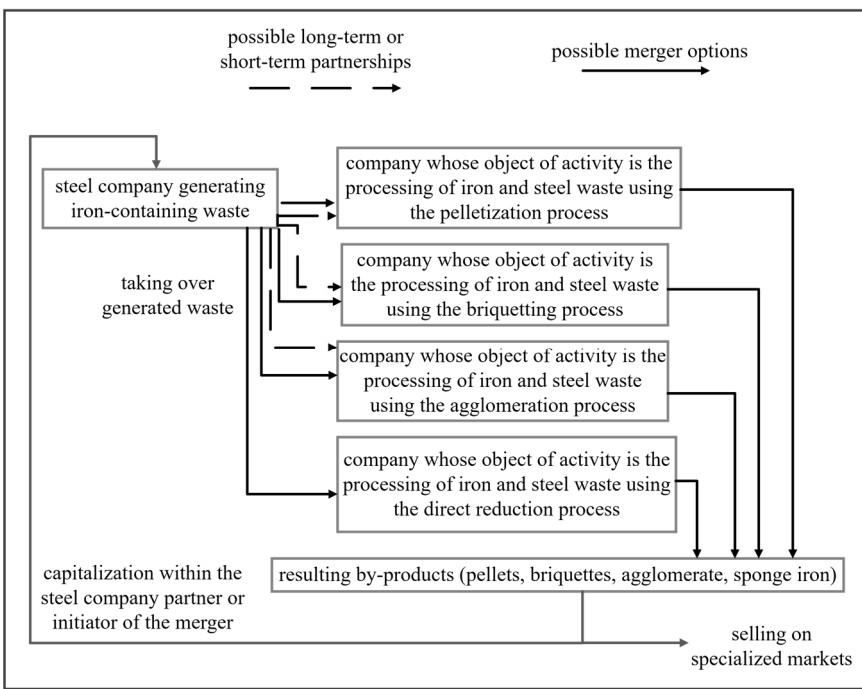

**Figure 10.** Conceptual model applicable to a joint-venture strategy.

The following limitations resulting from critical analysis must be considered: the risk of increasing environmental pollution through the transportation of waste from generators to processors and subsequently from processors to end users; the need for the continuous adaptation of chemical compositions for by-products, depending on the quality of the waste used. A modification of the inputs to the producer can automatically generate a modification of the composition of waste generated in the stream, as well as leading to a variation in the composition of historically stored waste through the sedimentation of heavier compounds.

The advantages of applying the conceptual model developed for the joint-venture strategy may include penetrating new markets, diversifying the range of by-products obtained from generated waste, ensuring a portion of the raw materials and materials required from internal sources, achieving a high level of sustainability in the field, as well as the opportunity for continuous development and growth for each partner company, providing valuable by-products for the steel industry or other industries.

The main disadvantages of implementing the conceptual model developed for the joint-venture strategy type can be highlighted through aspects related to potential conflicts or misunderstandings between partners, the duration of the partnership, difficulties encountered in the process of transferring waste and by-products to the main beneficiaries, difficulties in entering international markets due to competitive environment, and limited quantities of by-products that can be offered for sale or used as raw materials within the initiating steel company's partnerships.

Another approach to designing a concentration strategy involves the option of product development, which includes qualitative improvements to the core product or service, or the addition of a new analogous product or service [57]. In the field of steel industry, the market dictates the direction of production and/or the development of new brands of cast iron or steel. By restricting the base of traditionally used raw materials in the metallurgical field, including due to the application of globally assumed sustainable development concepts, the limitations resulting from the life cycle analysis of (ferrous) metallurgical products have been identified for the product development strategy—a longer product lifespan—leading to a reduction in the quantities of ferrous waste available for reuse [58].

### 4.3. Development of the Conceptual Model for Organizational Vertical Integration Strategy

The last option that can form the basis of designing a concentration strategy is horizontal integration, which involves the acquisition of companies that produce products or services similar to those of the acquiring company [57]. The limitations of the strategy are determined by the presence or absence of economic stability for the acquiring companies, which would allow for the acquisition of smaller firms, and the time horizon required for the implementation of new solutions in industrial practice. In the steel industry, this practice of assimilating other waste—producing or waste-processing steel companies—is not often opted for, with major industry players choosing to open their own sections for waste processing/valorization.

We also mention that it is possible to apply a vertical integration strategy, which involves expanding a company's operations into related business areas previously exploited by suppliers or customers of the firm (see Figure 11). A society that operates in the steel industry can begin to valorize industrial steel waste, becoming the main supplier for its own needs and other companies from the industry.

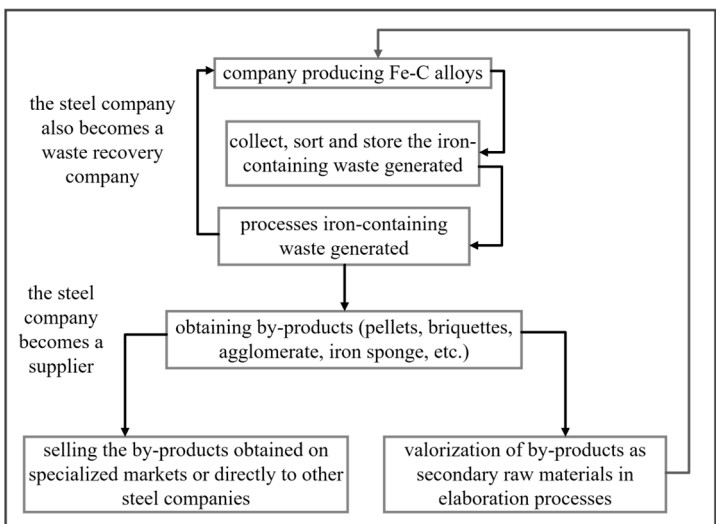

**Figure 11.** Conceptual model of an organizational strategy for vertical integration.

The advantages that can be identified in the event of applying the conceptual model of the organizational vertical integration strategy within a steel company are related to the following factors: improving the level of sustainable development in the steel industry by applying closed-loop concepts, highlighted by the valorization of its own waste in manufacturing cycles; contributing to the diversification of the existing raw material and material base; obtaining by-products that can successfully replace, to some extent, basic raw materials; diversifying the company's activities by acquiring the status of supplier for itself as well as for other companies in the industry.

The disadvantages that can be highlighted in the case of applying the conceptual model related to the organizational strategy of vertical integration are as follows: the level of market competition exerted by small companies processing steel waste; the variation in chemical and particle size compositions of waste, affecting the choice of optimal processing technology; inability to enter the waste valorization market due to low or variable quality of manufactured by-products compared to those of other patent-holding companies; a lack of information and data regarding the state of the relevant markets.

### 4.4. Developing the Conceptual Model for an Organizational Diversification Strategy

Another type of organizational strategy considered is that of diversification, which involves supporting the conduction of a series of activities within domains of activity that differ from the current domain of activity. It is considered necessary to mention that when

the diversification strategy involves entering a new field of activity that is different from the current one—but which corresponds with it in terms of marketed products, the markets in which it operates, or the technological procedures used—a concentric diversification organizational strategy will be practiced [57].

Figure 12 presents the conceptual model of an organizational diversification strategy, which is characteristic of companies operating in the steel industry.

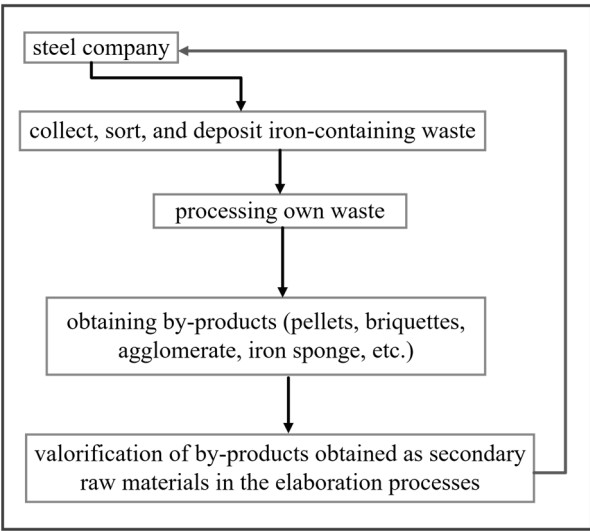

**Figure 12.** Conceptual model of organizational diversification strategy.

A steel company producing metal alloys such as steel and cast iron expands its activity and enters a related field of activity (waste valorization), thus ensuring its own raw material needs for the main activity (elaboration of Fe–C alloys) through its own forces; thus, it diversifies its own base of secondary raw materials, consisting of by-products obtained from processing the generated waste.

The potential advantages regarding the application of the conceptual model of an organizational diversification strategy are highlighted from the perspective of aspects related to the sustainability of a company: the transformation of its own generated waste into secondary materials for its own practiced elaboration processes; the necessary investments are minimal; the company most likely opts for classical procedures such as briquetting and pelletization, reducing the required amount of ferrous waste whose origin is uncertain by replacing the necessary quantities with their own by-products (pellets, briquettes).

The disadvantages that may arise in the case of applying the conceptual model of an organizational diversification strategy are primarily caused by the following factors: the financial situation of a company and its inability to acquire the necessary facilities for waste processing; the prolonged duration of time required to identify the optimal method for obtaining by-products once the processing technology is chosen; multiple attempts to improve and standardize the quality of the manufactured by-products.

## 5. Conclusions

The current context, both economically and geopolitically, is highlighted by frequent changes, with a rather turbulent environment. For these reasons, consumption needs and, implicitly, production needs undergo changes that managers must consider when designing and implementing strategies at their own companies' level.

With increased demand for steel products to support the growing needs in industrial sectors, such as construction, automotive, and infrastructure, and considering the objectives of sustainable development, solutions for the generation of secondary raw materials need to be found.

In this respect, in the present paper, we present our own research on pelletization of iron-containing waste in laboratory conditions. The obtained results recommend the

industrial implementation of the proposed recipes, which is evidenced by the metallic aspect of the pellets and, therefore, by their resistance to dropping.

Extensive research in the field of management, processing, and recovery of industrial steel waste with a high content of useful elements (Fe, C) was also presented; this resulted in the development of conceptual models of organizational strategies, which, in turn, represent proposals for improving/streamlining the process of the integrated management of these types of waste, applicable in steel-generating companies.

Through the conceptual models—which were drawn up and presented graphically—the authors sought to propose solutions centered on the management activities practiced within steel companies, emphasizing the importance of applying the elements of strategic management and the notion of organizational strategy. The proposed solutions were concentration, joint venture, vertical integration, and diversification.

It is imperative to identify and subsequently implement strategies through which interested companies can generate economic growth. If there is no change in the mentality of production or responsible consumption, there may be delays or even restructuring of production capacities. On the other hand, society has a moral obligation to identify possibilities to reduce and reuse the quantities of waste generated on current streams, to recover waste historically generated and currently landfilled, so that, by generating new secondary materials, we can meet future consumption needs.

**Author Contributions:** Conceptualization, I.F., G.P. and E.A.; methodology, I.F. and G.P.; validation, I.F., A.S., E.A. and M.A.; formal analysis, I.F., E.A. and M.A.; investigation, I.F., E.A. and A.S.; writing—original draft preparation, I.F. and G.P.; writing—I.F. and E.A.; All authors have read and agreed to the published version of the manuscript.

**Funding:** This research was funded by the Politehnica University Timisoara, through the program "Increasing the quality of advanced scientific research in UPT", according to the Decision of the Administrative Council no. 74/1/12.04.2022.

**Institutional Review Board Statement:** Not applicable.

**Informed Consent Statement:** Not applicable.

**Data Availability Statement:** The data presented in this study are available on request from the corresponding author.

**Acknowledgments:** This paper was financially supported by Politehnica University Timisoara, from its own income.

**Conflicts of Interest:** The authors declare no conflicts of interest. The founders had no role in the design of this study; in the collection, analyses, or interpretation of data; in the writing of the manuscript, or in the decision to publish the results.

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
