# Peer review of "Development of a Waste Management Strategy in a Steel Company"

_sustainability, doi:10.3390/su16114378_

Round 1
Reviewer 1 Report
Comments and Suggestions for Authors
In its current form, I believe the work is not suitable for publication. There is inconsistency between the different sections that constitute the work, it is not possible to clearly identify a methodology, as well as relevant results that contribute to increasing knowledge on the topic analysed. There are also many formal inaccuracies (e.g., the structure of the abstract, conclusions, introduction are incorrect) that make the work difficult to understand.
Comments on the Quality of English LanguageModerate editing of English language required
Reviewer 2 Report
Comments and Suggestions for Authors
This paper introduces the waste management in a steel company. The finding of this study benefits the utilization of waste. This paper can be considered after the following comment being addressed.
(1) The introduction section should be rewritten which is lacking a detailed introduction on the current us of the waste derived from the steel production. Besides, more of the paragraphs can be merged into one paragraphs, rather than one paragraph showing one reference result.
(2) The Figures. 5,7,8 and 9 can be removed from this paper, because the readers can
not obtain any necessary results and findings from these figures. Besides, more data and results in the literatures should be given and compared.
(3) Figure. 11 show no meaning at the current form.
(4) The steel waste can be recycled into the raw materials in the concrete. The author can give some introduction on the potential of the recycling of waste as recycled building materials. The following two reference may be helpful: (a) Reusing waste clay brick powder for low-carbon cement concrete and alkali-activated concrete: A critical review. (b) Characterization of sustainable mortar containing high-quality recycled manufactured sand crushed from recycled coarse aggregate.
(5) The cited reference should be removed from the conclusion section, in which the authors should highlight their findings.
Comments on the Quality of English LanguageModerate editing of English language required
Reviewer 3 Report
Comments and Suggestions for Authors
This work proposed for publication in Sustainability presents a possible strategy for recovery of steel-manufacturing-derived-waste based on the implementation of an improvement of existing technologies (direct reduction) and the management strategy.
The paper doesn’t follow the typical structure of a scientific work; however, I find the paper organization appropriate to successfully explain its content. It is not easy to couple experimental evidence with management strategy and Authors managed to make this argument understandable for readers. The shortcoming of this work is that neither the experimental part, nor the management section is deeply discussed. However, again I have to admit that going too deeply into both arguments would have been resulted in a paper not coherent, therefore Authors managed to keep a good balance.
I have a few recommendations prior to acceptance:
Abstract. The abstract has to be rewritten because it just provides a sort of summary of the section of the paper, but no real information on the content of the paper is given here. No key results, or key findings are provided. Since there are not highlights in Sustainability Journal, I suggest to reformulate the abstract to provide key finding of the works.
Materials and methods. This section contains also useful technology and process review. This section could be renominated as “technology overview” and indicate as “Materials and Methods” only the text referring to the experimental activity presented by authors (section 2.2).
Section 3. proposed solution. This section should also better explain the method employed to define the strategies further proposed in the text.
I also recommend to better explain or put a precise definition of “concentration strategy” introduced in section 3.1.
Reviewer 4 Report
Comments and Suggestions for Authors
In the work "Development of a Waste Management Strategy in a Steel Company", the main focus was on the regeneration of types of steel production waste, applying the circular economy concept.
The aim and hypothesis of the study are not emphasized in the abstract.
The work is described as providing a comprehensive analysis of the literature on the conversion of iron-containing waste into by-products for use in the steel industry. Researches were also carried out in laboratory conditions - using the granulation method of powdered waste. Laboratory research helped develop conceptual models of organizational strategies.
The study reveals that the main patterns of organizational strategies are concentration strategy, diversification, vertical integration, etc.
Graphical versions of the proposed strategies are developed in the work.
The introductory section provides an overview of the importance of circularity for the sustainable development of the steel industry under EU law.
The article examines the measures of the European Union (EU) to promote a sustainable and circular economy,
The world's most commonly used iron ore reduction processes are described in detail.
As highlighted in the article, the main goal of the Romanian specialist group is to create optimal technological flows for the recovery of black steel waste in a rotary kiln direct reduction plant.
The methodology section does not precisely name the scientific methods used in the research work.
The use of both theoretical and empirical research methods is not described.
A lot of verbiage.
I recommend numbering the conclusions.
Comments on the Quality of English LanguageA little editing of the English language is needed, which does not reduce the overall quality of the work.
Round 2
Reviewer 1 Report
Comments and Suggestions for Authors
The revised version of the paper is still not suitable for publication. Minor changes implemented by the Authors did not overcome lacks in the papers identified in the first revision round
Comments on the Quality of English LanguageModerate editing of English language required
Reviewer 2 Report
Comments and Suggestions for Authors
The author has addressed all my comments, and it can be published at the current form.
Reviewer 3 Report
Comments and Suggestions for Authors
I thank Authors for addressing my comments.
